# Post-acute sequelae of COVID-19 symptom phenotypes and therapeutic strategies: A prospective, observational study

Jennifer A. Frontera[1]*, Lorna E. Thorpe[2], Naomi M. Simon[3], Adam de Havenon[4], Shadi Yaghi[5], Sakinah B. Sabadia[1], Dixon Yang[6], Ariane Lewis[1], Kara Melmed[1], Laura J. Balcer[1,7], Thomas Wisniewski[1,8], Steven L. Galetta[1,7]

1 Department of Neurology, New York University Grossman School of Medicine, New York, New York, United States of America, 2 Department of Population Health, New York University, New York, New York, United States of America, 3 Department of Psychiatry, New York University Grossman School of Medicine, New York, New York, United States of America, 4 Department of Neurology, Yale University School of Medicine, New Haven, Connecticut, United States of America, 5 Department of Neurology, Brown University School of Medicine, Providence, Rhode Island, United States of America, 6 Department of Neurology, New York Presbyterian, Columbia Medical Center, New York, New York, United States of America, 7 Department of Ophthalmology, New York University Grossman School of Medicine, New York, New York, United States of America, 8 Department of Pathology, New York University Grossman School of Medicine, New York, New York, United States of America

* jennifer.frontera@nyulangone.org

**Data Availability Statement:** All relevant data are within the manuscript and its Supporting information files.

## Abstract

### Background

Post-acute sequelae of COVID-19 (PASC) includes a heterogeneous group of patients with variable symptomatology, who may respond to different therapeutic interventions. Identifying phenotypes of PASC and therapeutic strategies for different subgroups would be a major step forward in management.

### Methods

In a prospective cohort study of patients hospitalized with COVID-19, 12-month symptoms and quantitative outcome metrics were collected. Unsupervised hierarchical cluster analyses were performed to identify patients with: (1) similar symptoms lasting ≥4 weeks after acute SARS-CoV-2 infection, and (2) similar therapeutic interventions. Logistic regression analyses were used to evaluate the association of these symptom and therapy clusters with quantitative 12-month outcome metrics (modified Rankin Scale, Barthel Index, NIH NeuroQoL).

### Results

Among 242 patients, 122 (50%) reported ≥1 PASC symptom (median 3, IQR 1–5) lasting a median of 12-months (range 1–15) post-COVID diagnosis. Cluster analysis generated three symptom groups: Cluster1 had few symptoms (most commonly headache); Cluster2 had many symptoms including high levels of anxiety and depression; and Cluster3 primarily included shortness of breath, headache and cognitive symptoms. Cluster1 received few

**Funding:** The author(s) received no specific funding for this work.

**Competing interests:** Potential competing interest: JAF receives funding for the following COVID-19-related grants: NIH/NINDS 3U24NS11384401S1, NIH/NHLBI 1OT2HL161847-01, NIH/NIA 3P30AG066512-01; LET, NMS and LJB receive funding for the following COVID-19-related grant: NIH/NHLBI 1OT2HL161847-01; and TW receives funding for the following COVID-19-related grant: NIH/NIA 3P30AG066512-01. AdH, SY, SS, DY, AL, KM, and SLG do not report any relevant conflicts of interest. This does not alter our adherence to PLOS ONE policies on sharing data and materials.

therapeutic interventions (OR 2.6, 95% CI 1.1–5.9), Cluster2 received several interventions, including antidepressants, anti-anxiety medications and psychological therapy (OR 15.7, 95% CI 4.1–59.7) and Cluster3 primarily received physical and occupational therapy (OR 3.1, 95%CI 1.3–7.1). The most severely affected patients (Symptom Cluster 2) had higher rates of disability (worse modified Rankin scores), worse NeuroQoL measures of anxiety, depression, fatigue and sleep disorder, and a higher number of stressors (all P<0.05). 100% of those who received a treatment strategy that included psychiatric therapies reported symptom improvement, compared to 97% who received primarily physical/occupational therapy, and 83% who received few interventions (P = 0.042).

## Conclusions

We identified three clinically relevant PASC symptom-based phenotypes, which received different therapeutic interventions with varying response rates. These data may be helpful in tailoring individual treatment programs.

## Introduction

Post-acute sequelae of COVID-19 (PASC) have been reported in 7–91% of patients following acute SARS-CoV-2 infection [1–10]. This wide prevalence range is related to heterogeneous definitions of PASC [11–13], differences in populations being assessed (e.g. severe versus mild-moderate versus asymptomatic COVID-19 patients), and types of symptoms or signs included in rate estimates. Because there is no current biological definition of PASC, many studies lump disparate symptoms and signs into the PASC diagnosis, without an assessment of clinical relevance. The resulting heterogeneity in PASC cohorts makes it difficult to assess treatment strategies, which are likely to vary depending on PASC symptomatology. Disaggregating PASC into meaningful clinical phenotypes may help development of future targeted therapies.

In this prospective study, we aimed to identify clinically important phenotypes of PASC in patients hospitalized 12-months prior with COVID-19 illness, examine associations of these phenotypes with quantitative measures of functional status and quality of life, and evaluate whether PASC clusters received different therapeutic interventions. We further aimed to determine the rates at which patients reported subjective improvement with a given treatment program. While there are cluster analyses evaluating symptoms in the acute phase of COVID-19 illness [14–17], and others focused on specific post-acute symptoms such as pulmonary complaints [18], brain fog [19], or neuropsychiatric symptoms [20], we chose to evaluate the entire post-acute COVID-19 symptom report form published by the World Health Organization (WHO) [21] and map symptom clusters to therapeutic interventions. These analyses may provide insight into the efficacy of various symptom-based treatment strategies and underscore the need for a multi-disciplinary holistic approach to post-COVID-19 care.

## Methods

### Study design and patient cohort

We conducted a prospective, observational outcome study of consecutive COVID-19 patients hospitalized at four New York City area hospitals within one academic system between March 10, 2020-May 20, 2020. Follow-up interviews were performed at 12-months (+/- 2 months)

after initial SARS-CoV-2 diagnosis (window: January 10, 2021-July 20, 2021). Detailed enrollment, methodology and outcome measures have been previously reported [2, 3, 22, 23]. Inclusion criteria were: RT-PCR positive SARS-CoV-2 infection, age ≥18 years, hospital admission, and consent to participate in a follow-up interview. Exclusion criteria were: evaluation in an outpatient or emergency department setting only. PASC was defined according to Centers for Disease Control and Prevention (CDC) criteria as new or persistent symptoms occurring ≥4 weeks after SARS-CoV-2 infection [11].

## Standard protocol approvals and patient consents

This study was approved by the NYU Grossman School of Medicine Institutional Review Board. All patients or their surrogates provided consent for participation. All patients or their surrogates provided verbal consent for participation. The IRB did not require written consent because follow-up visits were conducted via phone call due to social distancing guidelines in place during the pandemic. Additionally, this research was deemed low risk by the IRB and verbal consent was allowed due to the observational nature of the study. Consent was documented in REDCap.

## 12-month follow-up questionnaires

The post-COVID symptom interview was based on the clinical case report form for PASC developed by the World Health Organization (WHO)(21). Subjects were asked if they believed they had "post-COVID syndrome", "long-hauler syndrome" or "long-COVID" characterized by symptoms occurring ≥4 weeks after initial COVID diagnosis. If the response was affirmative, subjects were asked to identify their symptoms from a list of 31 options [21] (S1 Table). Similarly, subjects were asked "What treatments have you used to deal with your prolonged COVID symptoms?" and were asked to indicate if they had used any of 14 possible therapeutic interventions, and whether the therapy improved their post-COVID symptoms (S2 Table). Subjects were also asked to indicate if they had experienced any of 15 stressors [1] within the month prior to the 12-month interview (S3 Table).

In addition to the above patient-reported outcomes, the following standardized batteries were administered at the 12-month interview: the modified Rankin Scale (mRS; 0 = no symptoms, 6 = dead) [24], the Barthel Index of activities of daily living (0 = completely dependent, 100 = independent for all activities) [25], the telephone-MoCA (22 = perfect score; ≤18 = abnormal cognition) [26], and Quality of Life in Neurological Disorders [27] (Neuro-QoL) short form self-reported health measures of anxiety, depression, fatigue and sleep. NeuroQoL raw scores were converted into T-scores with a mean of 50 and standard deviation of 10 in a reference population (U.S. general population or clinical sample) [28]. Higher T-scores indicate worse self-reported health for the anxiety, depression, fatigue and sleep metrics.

## Statistical analyses

Unsupervised hierarchical agglomerative cluster analyses were performed using Ward's method to place patients into similar clusters suggested by the data, not into *a priori* pre-defined categories. Case similarity was measured using variance computed from a fourfold table as (b+c)/4n, where *b* and *c* represent the diagonal cells corresponding to cases present on one item but absent on the other, and *n* is the total number of observations. We selected unsupervised agglomerative hierarchical cluster analysis methods because they provided a flexible approach to assemble symptoms clusters (based on similarity of their taxonomic structures) without prior knowledge about any underlying relationships, *and* because they allowed us to generate easy-to-interpret views of the clustering structure, an advantage over some other

clustering approaches. Agglomerative approaches are also efficient when datasets are smaller, such as ours [29]. Unlike k-means and some other clustering approaches, this method does not require advanced selection of number of clusters, and smaller clusters tend to be generated, which was useful in our exploration of PASC sub-phenotypes. We used Ward's minimum variance methods to minimizes the total within-cluster variance.

For symptom clustering, all 31 binary symptom variables were entered into the model in the order of their prevalence. Similarly, for therapy cluster analysis, all 14 binary therapy variables were entered into the model in the order of most to least frequently used. No other variables were entered into either cluster analysis. The number of clusters was determined based on inspection of the dendogram. To ensure cluster stability, cluster analyses were repeated in random subsamples of patients. In sensitivity analyses, we repeated hierarchical cluster analysis evaluating only incident symptoms (e.g. brain fog/confusion/memory loss, headache, anxiety, depression and shortness of breath could only be coded as present if the patient did not have a pre-COVID history of these symptoms). Because cluster analysis is performed on datasets without missing variables, patients with missing data were removed. To ensure the complete data sample was representative of the full sample, we contrasted the dataset used in the cluster analysis and the dataset of individuals with missing data using Chi-square tests for categorical measures and Mann-Whitney-U tests for continuous measures.

Demographics, stressors, comorbidities, hospital metrics and 12-month quantitative metrics were compared between symptom clusters, as well as between therapy groups, using Kruskal-Wallis tests for continuous, non-normally distributed data and Chi-squared or Fisher's exact tests for dichotomous variables. The association of therapy groups with 12-month outcome metrics were assessed using binary logistic regression analyses. Multivariable logistic regression analyses were used to assess the relationship between each therapy cluster with the rate of subjective improvement with treatment, adjusting for severity of index COVID-19 illness (intubated vs. not-intubated). All analyses were conducted using IBM SPSS Statistics for Mac version 28 (IBM Corp., Armonk, NY).

## Results

### PASC symptoms and impact

Follow-up interviews 12-months after hospitalization for COVID-19 were attempted in 590 patients and completed in 242/590 (41%)(3). The median age was 65 (IQR 53–73), 64% were male, 81/242 (34%) required invasive mechanical ventilation during index hospitalization and 113/242 (47%) had a neurological complication during their hospitalization. Of these patients, 122/242 (50%) reported ≥1 prolonged COVID-19 symptom lasting a median of 12-months (range 1–15 months). Multiple symptoms were reported in 87/122 (71%), and the median number of symptoms was 3 (IQR 1–5). The most prevalent symptoms included shortness of breath in 73/122 (60%), headache in 54/122 (44%) and cognitive symptoms, including difficulty concentrating, "brain fog" and/or memory issues in 48/122 (39%, Fig 1A). PASC symptoms interfered with work activities in 55/122 (45%), household responsibilities in 61/122 (50%) and leisure activities in 66/122 (54%). There were no differences in age, sex, past medical history (hypertension, diabetes, lung disease, dementia, psychiatric illness, headache disorder), vaccination status or proportion of patients with neurological events when comparing patients with or without PASC symptoms (all P>0.05). However, the severity of index COVID-19 infection—as measured by requirement of invasive mechanical ventilation—was significantly associated with the development of PASC symptoms (73% of ventilated patients developed PASC compared to 43% of patients who were not mechanically ventilated, P<0.001). Of those who completed 12 month follow-up, data on post-acute symptoms was missing on 12 patients.

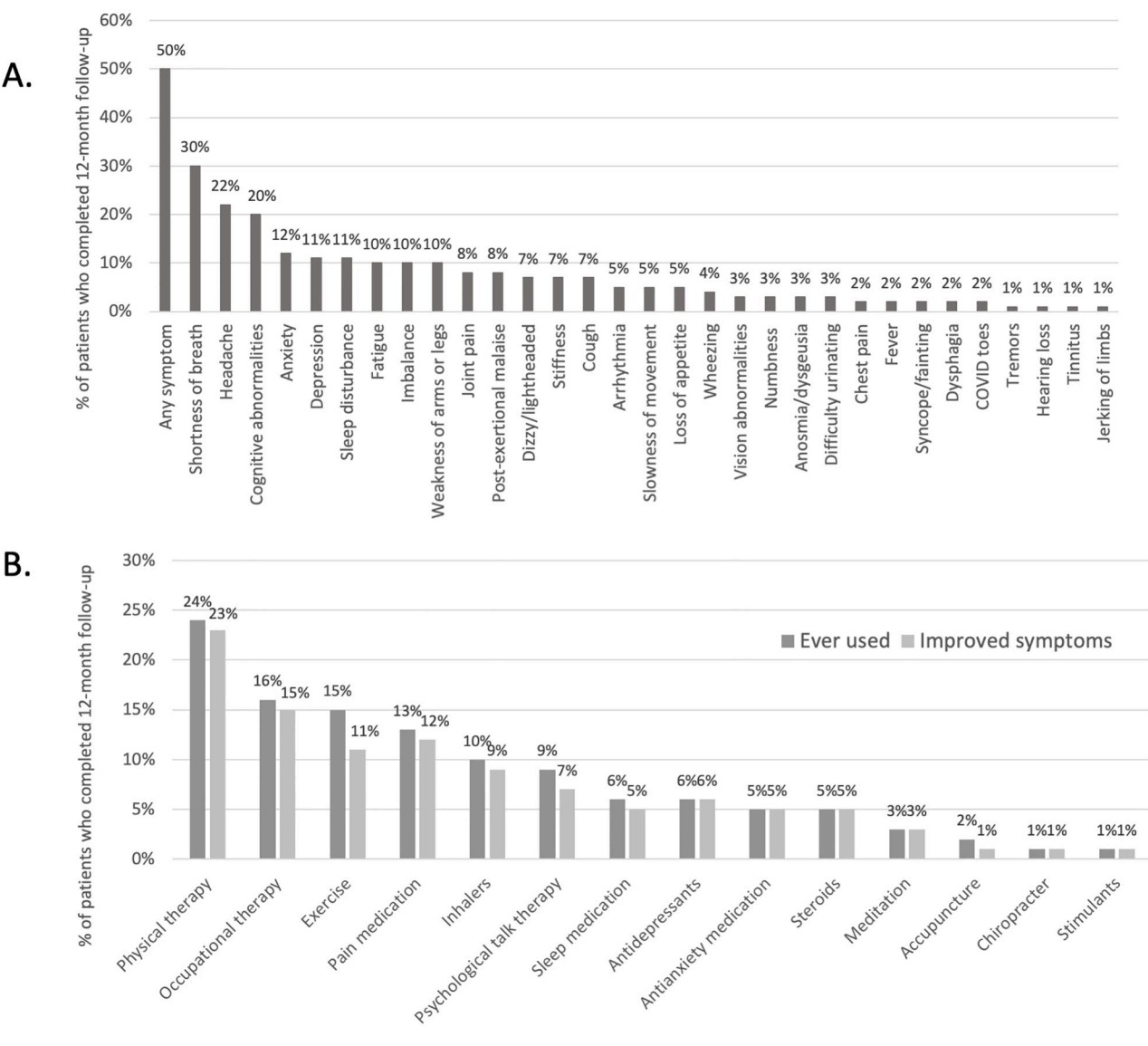

**Fig 1. Frequency of PASC symptoms, therapies received and rates of response to therapy. A.** Frequency of post-COVID symptoms lasting ≥ 4 weeks after initial diagnosis as reported by participants 12-months after hospitalization for COVID-19 (N = 242). **B.** Percent of patients who received, and responded to a given therapy for PASC symptoms within 12-months after hospitalization for COVID-19 (N = 242).

When comparing data among included patients (N = 122) to those with missing data (N = 12), those with missing data were significantly older (median 74 versus 64 years, P = 0.002), had fewer life stressors (median 0 versus 1, P = 0.010) and had worse 12-month modified Rankin scores (median 4 versus 2, P<0.001, S5 Table).

## PASC therapies and response rates

Of 122 PASC patients, 85 (70%) received at least one therapy to manage PASC symptoms and 72/122 (59%) received multiple therapies. The most frequently prescribed interventions were physical or occupational therapy, exercise or pain medications. Psychiatric interventions included: talk therapy (9%), antidepressants (6%) and anti-anxiety medications (5%). Most

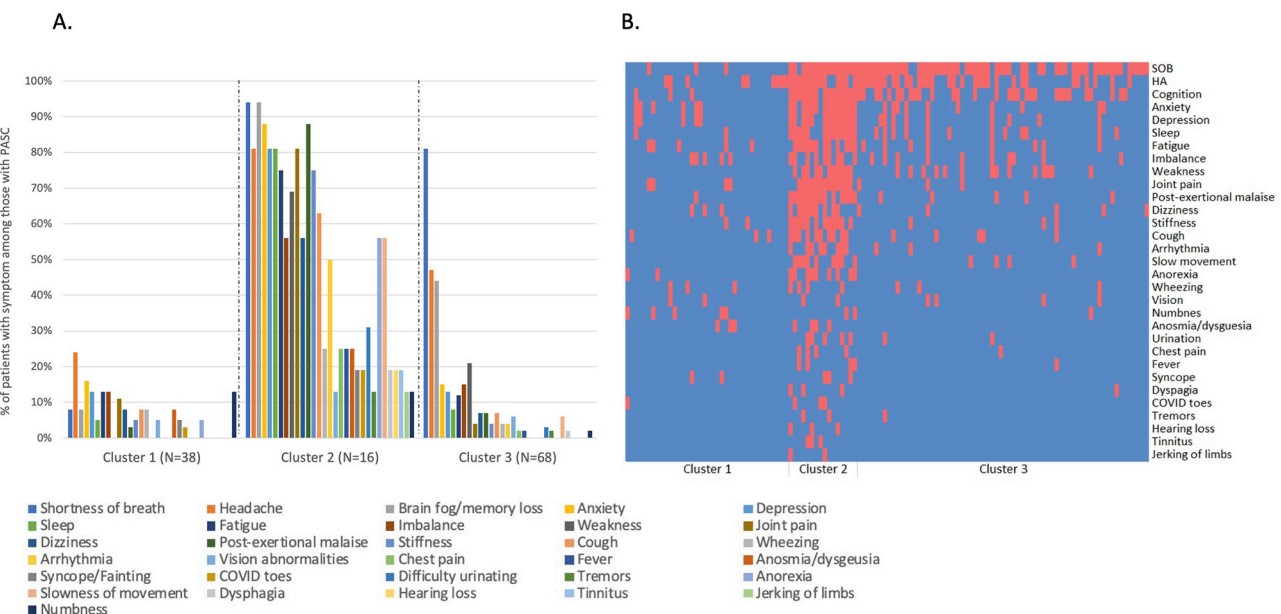

**Fig 2. Symptom clusters among patients with PASC (N = 122).** Cluster 1 has few symptoms; Cluster 2 has multiple symptoms with high rates of anxiety, depression and post-exertional malaise; Cluster 3 has high rates of shortness of breath, headache, and cognitive issues. Panel **A.** shows the frequency of each of the assessed symptoms in each cluster. Colors corresponding to each symptom are shown from left to right in the key. Panel **B.** shows a heat map for each patient, where red indicates presence of a symptom and blue indicates absence. SOB = shortness of breath; HA = headache.

patients reported a positive response to each therapy (range 75–100% reported symptom improvement), though a smaller proportion of patients (77%) felt that exercise alleviated their symptoms, and 7/85 (8%) reported no improvement with any intervention (Fig 1B).

## Symptom cluster analysis

Hierarchical cluster analysis generated three symptom clusters (Fig 2A and 2B): Symptom Cluster 1 (N = 38, 31%) included patients with few symptoms, most often headache and anxiety though at lower rates than reported in other clusters; Symptom Cluster 2 (N = 16, 13%) consisted of patients with many symptoms, including high levels of anxiety and depression; and Symptom Cluster 3 (N = 68, 56%) included patients who predominantly reported shortness of breath, headache and cognitive abnormalities. Sensitivity analyses evaluating only index symptoms post-COVID yielded similar cluster results to the primary analysis (S1 Fig). Symptom clusters did not differ significantly by age, sex, race, pre-COVID-19 disability status (baseline mRS) or medical comorbidities (Table 1). However, patients in Symptom Cluster 2 had the highest number of symptoms and stressors, were least likely to report that their symptoms did not affect routine activities, and had worse 12-month quantitative outcomes as compared to the other groups, including higher mRS scores (indicative of worse disability), as well as increased levels of anxiety, depression, fatigue and sleep abnormalities, as measured by NeuroQoL metrics (all P<0.05). In contrast, Symptom Cluster 1 had the fewest number of symptoms, less severe index COVID-19 illness (significantly lower rates of mechanical ventilation and hypotension), was most likely to report that symptoms did not impact routine activities, and had the lowest measures of anxiety, depression, sleep abnormalities and fatigue.

**Table 1. Demographics, stressors, co-morbidities, and 12-month metrics among symptoms clusters (N = 122 patients with PASC).**

| | Symptom Cluster 1 (N = 38) Few symptoms | Symptom Cluster 2 (N = 16) Many symptoms with high levels of anxiety and depression | Symptom Cluster 3 (N = 68) Predominantly shortness of breath, headache and cognitive issues | P across all 3 groups |
|---|---|---|---|---|
| **Demographics** | | | | |
| Age, median (IQR) | 62 (49–70) | 63 (52–71) | 64 (58–72) | 0.351 |
| Sex (male), N (%) | 25/38 (66%) | 7/16 (44%) | 43/68 (63%) | 0.285 |
| Race (white), N (%) | 19/32 (59%) | 9/16 (56%) | 33/51 (54%) | 0.791 |
| Education level >12 years, N (%) | 27/33 (82%) | 10/13 (77%) | 53/64 (83%) | 0.882 |
| **Stressors** | | | | |
| At least one stressor, N (%) | 22/38 (58%) | 14/16 (88%) | 38/68 (56%) | 0.061 |
| Number of stressors, median (IQR) | 1 (0–2) | 2 (1–5) | 1 (0–2) | **0.005** |
| Social Isolation, N (%) | 6/38 (16%) | 5/16 (31%) | 10/68 (15%) | 0.277 |
| Financial Insecurity, N (%) | 6/38 (16%) | 6/16 (38%) | 12/68 (18%) | 0.153 |
| Unemployment, N (%) | 5/38 (13%) | 3/16 (19%) | 9/68 (13%) | 0.837 |
| Food Insecurity, N (%) | 1/33 (3%) | 0/12 (0%) | 0/39 (0%) | 0.457 |
| Homelessness, N (%) | 0/38 (0%) | 0/16 (0%) | 1/68 (2%) | 0.670 |
| Domestic violence, N (%) | 0/38 (0%) | 0/16 (0%) | 0/68 (0%) | — |
| Relationship problems in household, N (%) | 2/38 (5%) | 2/16 (13%) | 5/68 (7%) | 0.649 |
| Education disruption, N (%) | 1/38 (3%) | 0/16 (0%) | 3/68 (4%) | 0.648 |
| Increased caregiver responsibilities, N (%) | 1/38 (3%) | 2/16 (13%) | 5/68 (7%) | 0.378 |
| Personal Illness, N (%) | 8/38 (21%) | 7/16 (44%) | 22/68 (32%) | 0.218 |
| New Disability, N (%) | 2/38 (5%) | 5/16 (31%) | 5/68 (7%) | **0.008** |
| Death of close contact, N (%) | 0/38 (0%) | 10/16 (63%) | 6/68 (9%) | **<0.001** |
| Illness of close contact, N (%) | 5/38 (13%) | 3/16 (19%) | 6/68 (9%) | 0.494 |
| Lack of access to child care, N (%) | 0/38 (0%) | 1/16 (6%) | 0/68 (0%) | **0.035** |
| Political conflict with close contacts, N (%) | 2/38 (5%) | 0/16 (0%) | 6/68 (9%) | 0.407 |
| **Comorbidities** | | | | |
| Pre-COVID disability (mRS), median (IQR) | 0 (0–0) | 0 (0–2) | 0 (0–1) | 0.488 |
| Hypertension, N (%) | 9/38 (24%) | 8/16 (50%) | 28/68 (41%) | 0.102 |
| Diabetes, N (%) | 5/38 (13%) | 5/16 (31%) | 22/68 (32%) | 0.087 |
| COPD/Asthma, N (%) | 3/38 (8%) | 4/16 (25%) | 7/68 (10%) | 0.178 |
| Headache Disorder, N (%) | 4/38 (11%) | 0/16 (0%) | 2/68 (3%) | 0.139 |
| Dementia, N (%) | 2/38 (5%) | 0/16 (0%) | 7/68 (10%) | 0.306 |
| Psychiatric history, N (%) | 3/38 (8%) | 2/16 (13%) | 6/67 (9%) | 0.864 |
| **Index COVID-19 Hospitalization** | | | | |
| Neuro complication, N (%) | 17/38 (45%) | 6/16 (38%) | 31/68 (46%) | 0.840 |
| Mechanically ventilated, N (%) | 10/38 (26%) | 8/16 (50%) | 39/68 (57%) | **0.009** |
| Worst Sequential Organ Failure Assessment (SOFA) score, median (IQR) | 3 (3–9) | 6 (3–13) | 6 (3–12) | 0.062 |
| Lowest % oxygen saturation, median (IQR) | 86% (76–92%) | 80% (68–87%) | 78% (66–88%) | 0.251 |
| Lowest mean arterial blood pressure (mmHg), median (IQR) | 70 (56–79) | 59 (55–68) | 60 (48–71) | **0.020** |
| Acute renal failure, N (%) | 2/38 (5%) | 4/16 (25%) | 15/68 (22%) | 0.061 |
| **Impact of Symptoms** | | | | |
| Interfered with work, N (%) | 13/38 (34%) | 9/16 (56%) | 33/68 (49%) | 0.229 |

*(Continued)*

**Table 1.** (Continued)

| | Symptom Cluster 1 (N = 38) Few symptoms | Symptom Cluster 2 (N = 16) Many symptoms with high levels of anxiety and depression | Symptom Cluster 3 (N = 68) Predominantly shortness of breath, headache and cognitive issues | P across all 3 groups |
|---|---|---|---|---|
| Interfered with household responsibilities, N (%) | 16/38 (42%) | 10/16 (63%) | 35/68 (52%) | 0.367 |
| Impacted leisure activities, N (%) | 15/38 (40%) | 10/16 (63%) | 41/68 (60%) | 0.092 |
| No impact of symptoms on routine activities, N (%) | 16/38 (42%) | 1/16 (6%) | 8/68 (12%) | **<0.001** |
| Duration of symptoms in months, median (IQR) | 12 (4–13) | 12 (12–12) | 12 (12–13) | 0.496 |
| **12-month Quantitative Metrics** | | | | |
| Number of symptoms, median (IQR) | 1 (1–3) | 16 (13–18) | 3 (2–5) | **<0.001** |
| 12-mo Barthel Index, median (IQR) | 100 (100–100) | 98 (68–100) | 100 (90–100) | 0.053 |
| 12-mo T-MoCA, median (IQR) | 19 (16–20) | 18 (14–19) | 18 (15–20) | 0.206 |
| 12-mo mRS, median (IQR) | 2 (0–3) | 3 (2–4) | 2 (2–4) | **0.047** |
| 12-mo NeuroQoL Anxiety, median T-score (IQR) | 47 (36–54) | 58 (52–62) | 50 (44–55) | **0.001** |
| 12-mo NeuroQoL Depression, median T-score (IQR) | 40 (37–50) | 50 (43–59) | 47 (37–51) | **0.047** |
| 12-mo NeuroQoL Fatigue, median T-score (IQR) | 44 (39–51) | 58 (48–62) | 50 (42–55) | **<0.001** |
| 12-mo NeuroQoL Sleep, median T-score (IQR) | 46 (33–55) | 57 (52–66) | 50 (40–56) | **0.003** |

## Therapy cluster analysis

A separate hierarchical cluster analysis was performed to identify groups of patients who received similar therapeutic interventions. Three clusters were generated (Fig 3A and 3B): Therapy Group A (N = 72, 59%) received few therapeutic interventions, most often pain medications and exercise; Therapy Group B (N = 12, 10%) received several interventions, notably psychological talk therapy, antidepressants, and anti-anxiety medications; and Therapy Group

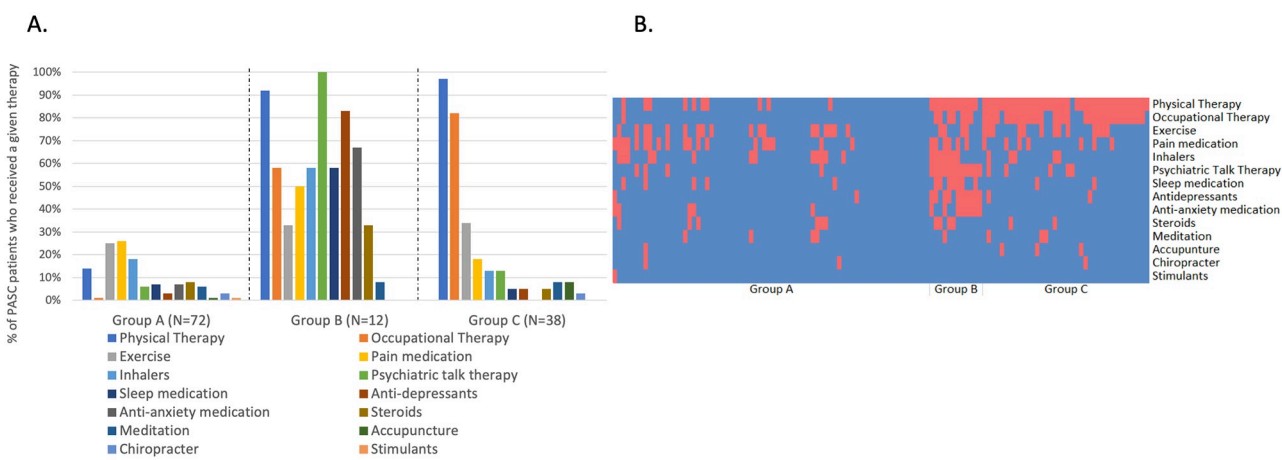

**Fig 3. Therapeutic groups among PASC patients (N = 122).** Group A received few therapeutic interventions; Group B received several interventions, most notably psychological talk therapy, anti-depressants, anti-anxiety medications and physical therapy; Group C primarily received physical and occupational therapy. Panel **A.** shows the percent of patients who received a given therapy stratified by therapy group. Colors corresponding to each therapy are shown from left to right in the key. Panel **B.** shows a heat map where red indicates a given therapy was received and blue indicates that a therapy was not utilized.

C (N = 38, 31%) primarily received physical and occupational therapy. Patients in Therapy Group A were significantly younger than the other groups, and were less likely to have been mechanically ventilated, while Therapy Group C had significantly higher rates of mechanical ventilation (Table 2). Among those who received no therapeutic interventions (N = 37), the median age was 62 (IQR 51–72), 67% were male and 30% required mechanical ventilation during hospitalization. There were no significant differences in age or sex among those who did or did not receive a therapeutic intervention. However, those who did not receive any therapies were less likely to have required mechanical ventilation during index COVID-19 than those who received some intervention (30% versus 54%, P = 0.018).

## Relationship of symptom and therapy clusters

Although generated independently, symptom clusters and therapeutic groups were closely aligned. Inclusion in Symptom Cluster 1 significantly predicted Therapy Group A (odds ratio [OR] 2.55, 95% confidence interval [CI] 1.10–5.89), Symptom Cluster 2 was associated with Therapy Group B (OR 15.70, 95% CI 4.14–59.70) and Symptom Cluster 3 was associated with Therapy Group C (OR 3.08, 95% CI 1.33–7.13, Fig 4, S4 Table).

## Subjective improvement among therapy groups

Among patients who received at least one therapeutic intervention (N = 85), those in Therapy Group B and C were significantly more likely to report improvement following therapy than Therapy Group A (100%, vs. 97% vs 83%, respectively, P = 0.042). This was observed despite the fact that Therapy Group A had significantly better measures of activities of daily living (Barthel Index) and less severe disability (mRS) than other groups. After adjusting for severity of index COVID-19 illness (intubation status) and age, Therapy Group A was associated with significantly lower rates of subjective improvement compared to other therapy groups (adjusted OR 0.07, 95% CI 0.01–0.68, P = 0.022). Conversely, belonging to Therapy Groups B or C (compared to A) was strongly associated with subjective improvement (aOR 14.5, 95% CI 1.5–142.9, P = 0.022), after adjusting for age and ventilator status. There was a significant association between the number of therapeutic interventions trialed and the subjective report of improvement following therapy (OR 3.00, 95% CI 1.22–7.35, P = 0.017).

## Discussion

In this prospective study, hierarchical cluster analysis identified three distinct groups of PASC patients, who differed in type and frequency of post-COVID symptoms. These symptom clusters were clinically significant since they were associated with quantitative 12-month outcome measures of disability and quality of life. An independent cluster analysis also identified three therapeutic groups, and symptom clusters intuitively and statistically mapped to these therapy clusters. The most severely affected cluster (Symptom Cluster 2 representing 13% of PASC patients) had more symptoms, higher rates of disruption of routine activities due to symptoms, higher disability and higher quantitative measures of anxiety and depression, but also reported some of the highest rates of subjective symptom improvement with a therapeutic strategy that included psychiatric treatments (Therapy Group B). Given the high rates of incident anxiety disorders, depression, stress and adjustment disorders reported post-COVID-19 [30], and the stigma attached to mental health treatment [31], this excellent response rate suggests that psychiatric interventions for PASC are important in appropriately identified individuals. Since care for PASC patients may be siloed among different subspecialties, vigilance and aggressive management of psychiatric symptoms are warranted.

**Table 2. Demographics, comorbidities, symptom clusters and 12-month outcome metrics among Therapy Groups (N = 122 PASC patients).**

| | Therapy group A: (N = 72) Received few therapeutic interventions | Therapy group B: (N = 12) Received several interventions, most notably psychological talk therapy, anti-depressants, anti-anxiety medications and physical therapy | Therapy group C: (N = 38) Primarily received physical and occupational therapy | P across all 3 groups |
|---|---|---|---|---|
| **Demographics** | | | | |
| Age, median (IQR) | 62 (50–71) | 68 (63–71) | 66 (60–72) | **0.035** |
| Sex (male), N (%) | 45/72 (63%) | 5/12 (42%) | 25/38 (66%) | 0.314 |
| Race (white), N (%) | 33/55 (60%) | 9/12 (75%) | 19/32 (59%) | 0.595 |
| Education level >12 years, N (%) | 9/51 (18%) | 1/8 (13%) | 3/29 (10%) | 0.356 |
| **Comorbidities** | | | | |
| Pre-COVID disability (mRS), N (%) | 0 (0–1) | 0 (0–0) | 0 (0–2) | 0.460 |
| Hypertension, N (%) | 23/72 (32%) | 5/12 (42%) | 17/38 (45%) | 0.391 |
| Diabetes, N (%) | 16/72 (22%) | 3/12 (25%) | 13/38 (34%) | 0.395 |
| COPD/Asthma, N (%) | 9/72 (13%) | 2/12 (17%) | 3/38 (8%) | 0.647 |
| Headache disorder, N (%) | 5/72 (7%) | 1/12 (8%) | 0/38 (0%) | 0.235 |
| Dementia, N (%) | 6/72 (8% | 0/12 (0%) | 3/38 (10%) | 0.587 |
| Psychiatric history, N (%) | 4/71 (6%) | 3/12 (25%) | 4/38 (11%) | 0.091 |
| Neuro complication during index COVID-19 hospitalization, N (%) | 32/72 (44%) | 5/12 (42%) | 17/38 (45%) | 0.982 |
| Mechanically ventilated during hospitalization for COVID-19, N (%) | 24/72 (33%) | 7/12 (58%) | 26/38 (68%) | **0.001** |
| COVID-19 Vaccination, N (%) | 46/70 (64%) | 9/12 (75%) | 27/38 (71%) | 0.754 |
| **Symptom Clusters** | | | | |
| Symptom Cluster 1, N (%) | 24/34 (71%) | 0/12 (0%) | 9/38 (24%) | 0.052 |
| Symptom Cluster 2, N (%) | 1/34 (3%) | 10/12 (83%) | 1/38 (3%) | **<0.001** |
| Symptom Cluster 3, N (%) | 9/34 (27%) | 2/12 (17%) | 28/38 (74%) | **0.015** |
| Symptom Cluster 2 or 3 versus 1, N (%) | 44/72 (61%) | 11/12 (92%) | 29/38 (76%) | 0.052 |
| Number of PASC symptoms, N (%) | 2 (1–5) | 12 (4–18) | 3 (2–5) | **<0.001** |
| Number of therapies received, median (IQR) | 0 (0–2) | 7 (5–8) | 3 (2–4) | **<0.001** |
| Duration of symptoms, median (IQR) | 12 (10–12) | 12 (12–13) | 12 (12–13) | 0.087 |
| **12-month Outcomes** | | | | |
| Improved with therapy*, N (%) | 29/35 (83%) | 12/12 (100%) | 37/38 (97%) | **0.042** |
| 12-mo Barthel Index, median (IQR) | 100 (100–100) | 98 (61–100) | 98 (68–100) | **<0.001** |
| 12-mo T-MoCA, median (IQR) | 18 (15–20) | 18 (14–21) | 19 (17–20) | 0.480 |
| 12-mo mRS, median (IQR) | 2 (1–3) | 3 (1–4) | 3 (2–4) | **0.003** |
| 12-mo NeuroQoL Anxiety, median (IQR) | 50 (44–56) | 55 (39–63) | 48 (36–53) | 0.338 |
| 12-mo NeuroQoL Depression, median (IQR) | 45 (37–51) | 47 (38–57) | 47 (37–51) | 0.644 |
| 12-mo NeuroQoL Fatigue, median (IQR) | 48 (42–54) | 55 (45–60) | 48 (42–54) | 0.244 |

*(Continued)*

**Table 2.** (Continued)

| | Therapy group A: (N = 72) Received few therapeutic interventions | Therapy group B: (N = 12) Received several interventions, most notably psychological talk therapy, anti-depressants, anti-anxiety medications and physical therapy | Therapy group C: (N = 38) Primarily received physical and occupational therapy | P across all 3 groups |
|---|---|---|---|---|
| 12-mo NeuroQoL Sleep, median (IQR) | 52 (39–57) | 51 (46–65) | 46 (39–53) | 0.195 |

*among those who had symptoms and received at least one intervention, logistic regression analyses adjusted for severity of index COVID-19 (as assessed by requirement for invasive mechanical ventilation)

Bold = significance with P<0.05. mRS = modified Rankin Scale; T-MoCA = telephone Montreal Cognitive Assessment, NeuroQoL = NIH Neurological Quality of Life patient reported outcomes.

This study differs from other COVID-19 cluster analyses [15–20] in that, not only did we identify phenotypes of PASC symptoms, but we also mapped these clusters to corresponding therapeutic strategies and their subjective effectiveness. Disaggregating PASC, which is a highly heterogeneous condition, and identifying useful treatment strategies is critically needed, particularly since we found that 50% of hospitalized COVID-19 patients have PASC symptoms. The Centers for Disease Control and Prevention report 4,575,344 hospital admissions for COVID-19 since August 1, 2020 (which does not include the first wave in New York City) through February 2022 [32]. This would translate into approximately 2.3 million Americans who have had PASC symptoms within 12-months of hospitalization for COVID-19, in addition to the millions more who have PASC after mild-moderate COVID-19. Additionally, the economic and societal impact of PASC symptoms after hospitalization for COVID-19 is likely

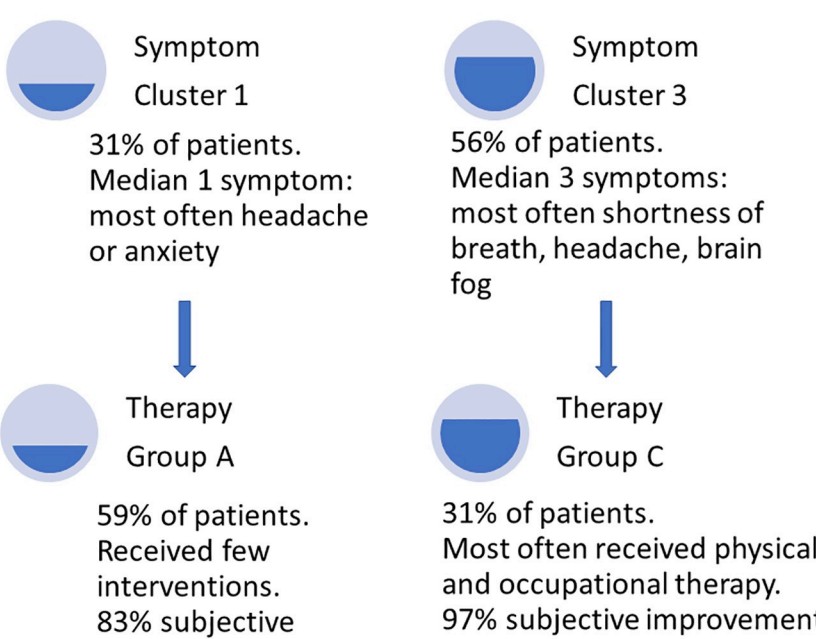
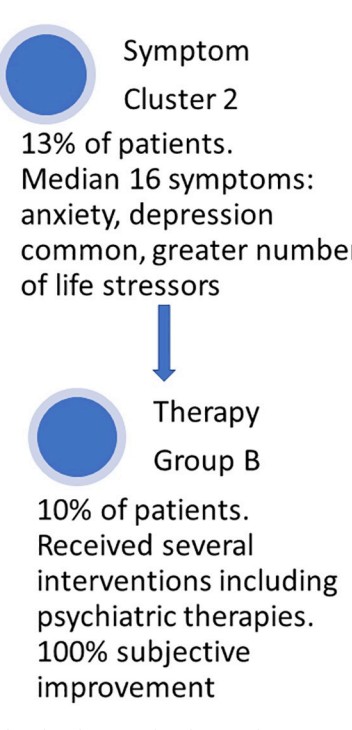

**Fig 4. Relationship of symptom clusters and therapy groups (N = 122).** Symptom Cluster 1 was significantly related to membership in Therapy Group A (Odds Ratio [OR] 2.55, 95% confidence interval [CI] 1.10–5.89), Symptom Cluster 2 was associated with Therapy Group B (OR 15.70, 95% CI 4.14–59.70) and Symptom Cluster 3 was associated with Therapy Group C (OR 3.08, 95% CI 1.33–7.13).

to be considerable since 80% of PASC patients in our study reported disruptions in their ability to work, manage household responsibilities or engage in leisure activities due to symptoms. Other strengths of this study include its prospective design, highly detailed characterization of the cohort from hospital stay through 12-month follow-up, and combination of patient reported outcomes and objective measures of functional status (mRS, Barthel Index, NeuroQoL).

We found that COVID-19 severity, even amongst a hospitalized cohort, was an overall predictor of PASC in the group of 242 patients that completed follow-up, as has been described in multiple other cohorts [33–35]. Although some studies have suggested that PASC may be more common in those with mild disease [36], we did not have a comparison group of non-hospitalized patients in our cohort to test this hypothesis. Among the subgroup of patients with PASC, the severity of index COVID-19 illness did not appear to predict which patients would have the most symptoms or disability. Indeed, the most severely affected group (Symptom Cluster 2) had not only the highest number of symptoms, but also worse measures of functional outcome, even though this group had less severe index COVID-19 illness (based on intubation rates), and similar rates of neurological complications compared to other clusters. Our findings imply that there are other factors that impact PASC severity. Notably, patients in Symptom Cluster 2 reported a significantly higher number of life stressors than other clusters. In a prior study of risk factors for PASC among U.S. community dwellers with and without mild COVID-19 conducted in February 2021, we identified multiple stressors (present within the month prior to interview) that were associated with the development of PASC, most notably financial insecurity and unemployment [1]. In that study, multivariable models predicting NeuroQoL measures of cognition, anxiety, depression, fatigue and sleep, demonstrated that several stressors were stronger predictors of abnormalities on quality of life testing than was COVID-19 infection itself. These data suggest an interplay of environmental and pandemic-related factors that may impact the development of PASC.

A positive finding of this paper is that most patients who received a given therapy or group of therapies reported improvement, and only 8% reported no improvement at all with any trialed intervention. These data are encouraging, particularly since no patient received a therapy specifically engineered to target PASC symptoms. Practitioners appeared to prescribe therapies directed towards symptoms, and were largely successful with this approach. However, not all patients reported the same rate of symptom improvement with a given therapeutic strategy. Notably, the patients in Therapy Group A, who received the fewest therapeutic interventions, and were the youngest and least disabled (based on mRS and Barthel Index scores), were also the least likely to report improvement following therapy. These data suggest that these patients may have had relatively mild PASC and small improvements were difficult to detect, or that too few or inadequate interventions were trialed. We did identify a significant association between the number of interventions used and the reported rates of symptomatic improvement. Whether this relationship is merely related to trial and error resulting in selection of effective treatments, increased interaction with the healthcare system in those that received more therapies, or other factors such as PASC severity, remains to be determined. Of the 14 interventions tested, exercise appeared to be the least likely to lead to symptomatic relief. This is noteworthy because of the similarities PASC shares with chronic fatigue syndrome, in which symptoms are characteristically worsened with exertion [37]. However, only 16% of PASC patients in our study reported post-exertional malaise, suggesting that phenotypic similarities with chronic fatigue syndrome may only occur in a subset of PASC patients.

There are several limitations to this study. First, the study population had COVID-19 requiring hospitalization and results of our study may not generalize to non-hospitalized COVID-19 patients. However, the types and prevalence of PASC symptoms we reported are

similar to those described following mild-moderate COVID-19 [1, 35, 38–40]. Second, data was missing in 12 patients who were excluded from cluster analyses. Those excluded from analyses were largely similar to those included, however excluded patients were significantly older, had worse 12-month mRS scores, and had fewer life stressors than those included. Additionally, we were only able to contact 242/590 (41%) patients eligible for 12-month follow-up due to defunct contact information, language barriers, lack of patient/surrogate consent or unanswered phone calls. Though we made three phone attempts at contact, our data may represent a biased sample. Third, because this was not a controlled, clinical trial, we are unable to determine if one therapeutic intervention was superior to another. Indeed, since the majority of patients received multiple treatments and these treatments may augment or interact with one another, the only way to deal with this heterogeneity was with cluster analysis. Additionally, there were too few patients in certain treatment categories to make any statements regarding efficacy. While those in Therapy Group A reported lower rates of subjective improvement compared to Groups B or C, confidence intervals were wide suggesting imprecision in point estimates. Fourth, though patients reported subjective improvement with therapy, we do not have objective measures of improvement, and we cannot exclude the possibility that responses to interventions may represent a "placebo effect". Fifth, we did not have detailed information on the severity of symptoms or the degree of improvement with therapy, since these data were collected in a dichotomous fashion. It is possible that those in Therapy Group A who reported lower rates of symptomatic improvement with therapy had less severe symptoms, and hence, less room for dramatic improvements with intervention. Conversely, those with several symptoms of varying severity might require only modest interventions to notice results. We attempted to mitigate against this bias by adjusting for initial severity of COVID-19 illness, and we still demonstrated a lower likelihood of subjective improvement in Therapy Group A. Finally, the spectrum of therapies we assessed was wide, but not comprehensive. For example, we did not evaluate cognitive therapies, and we did not have details regarding which psychiatric medications were most efficacious. Future studies focusing on specific therapeutic interventions are warranted.

## Conclusions

PASC following severe COVID-19 is a heterogeneous condition that can be phenotyped into three clinically relevant symptom clusters. These symptom clusters differed in number and impact of symptoms, disability status and quantitative measures of anxiety, depression, sleep and fatigue, and were significantly associated with independently generated therapy clusters. However, response rates to therapeutic strategies differed significantly among groups. Matching PASC symptom phenotypes with corresponding therapeutic strategies may be helpful in tailoring individual treatment programs. Importantly, identifying the most severely affected PASC patients (i.e. Symptom Cluster 2) and implementing psychiatric treatments should be considered in clinical practice.

## Supporting information

**S1 Table. Symptom questionnaire.** Subjects were asked 12-months after hospitalization for COVID, "Do you believe you have "post-COVID syndrome" also called "long hauler" syndrome or did you continue to have symptoms for at least 4 weeks after your initial diagnosis of COVID-19?" If the response was "yes", they were presented with the options shown in this table.
(DOCX)

**S2 Table. Therapy questionnaire.** Subjects were asked, "What treatments have you used to deal with your prolonged COVID symptoms? Please check left column if ever used, and also right column if therapy helped post-COVID symptoms".
(DOCX)

**S3 Table. Stressor questionnaire.** Subjects were asked "Have you experienced any of the following stressors within the last month? (check all that apply in the past month)".
(DOCX)

**S4 Table. Univariate odds ratios and 95% confidence intervals representing the association of demographics, comorbidities, symptom clusters and 12-month outcome metrics with therapy groups (N = 122 PASC patients).**
(DOCX)

**S5 Table. Comparison of patients included in analyses to patients with missing data.**
(DOCX)

**S1 Fig. Sensitivity analysis of incident symptom clusters using hierarchical cluster analysis (Ward's method, variance measurement).** The three symptom clusters generated are similar to the primary analysis.
(TIFF)

## Acknowledgments

We would like to thank the patients and families that participated in this study.

## Author Contributions

**Conceptualization:** Jennifer A. Frontera, Steven L. Galetta.

**Data curation:** Jennifer A. Frontera, Sakinah B. Sabadia, Dixon Yang, Kara Melmed.

**Formal analysis:** Jennifer A. Frontera, Lorna E. Thorpe.

**Investigation:** Dixon Yang.

**Methodology:** Jennifer A. Frontera, Naomi M. Simon, Adam de Havenon, Shadi Yaghi, Ariane Lewis.

**Resources:** Sakinah B. Sabadia.

**Supervision:** Laura J. Balcer, Thomas Wisniewski, Steven L. Galetta.

**Writing – original draft:** Jennifer A. Frontera.

**Writing – review & editing:** Jennifer A. Frontera, Lorna E. Thorpe, Naomi M. Simon, Adam de Havenon, Shadi Yaghi, Ariane Lewis, Kara Melmed, Laura J. Balcer, Thomas Wisniewski, Steven L. Galetta.

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
