## [Decision Letter · Decision Letter 0]

7 Jul 2022

PONE-D-22-12045Post-Acute Sequelae of COVID-19 Symptom Phenotypes and Therapeutic Strategies:  A Prospective, Observational Study PLOS ONE

Dear Dr. Frontera,

Thank you for submitting your manuscript to PLOS ONE. After careful consideration, we feel that it has merit but does not fully meet PLOS ONE’s publication criteria as it currently stands. Therefore, we invite you to submit a revised version of the manuscript that addresses the points raised during the review process.

Please make sure to incorporate the following as suggested by the reviewers and editor (in addition): Add some description of cluster 1 symptom rates in the results section based on Fig 2A (and suppl Fig 1, also see Reviewer 1 comment)Please redo Table 2 completely, because there are a lot of redundant data included (also see Reviewer 2 comment):The N for the denominator can be given when different from given in the header or provide N (%) with missing data for applicable subsets.Would it make sense to assess differences in variable distributions among the Therapy groups using univariable analysis (chi squared tests and nonparametric/anova test for continuous variables)?For the symptom clusters: it may be interesting to compare odds of falling within cluster 2 and 3 vs cluster 1 (reference group) among Therapy groups.For the 12-month outcomes, e.g., "improved with therapy", it may be interesting to compare Therapy group B and C vs A, while adjusting for cluster 2 and 3 dummy variables as well as severity of disease + testing for interaction between the therapy dummy variables (Therapy group B and C) and cluster dummy variables. Perhaps these analyses based on above 2 suggestions could be added separately.Were continuous variables (incl age, continuous 12-mo outcomes) analyzed as binary outcomes with the median as cut-off value? No wonder the OR are close to 1 when there are sufficient data to estimate the median, and when Ns above vs below median are compared in each cell (i.e., 0.5/(1-0.5)=1). Please include results as supplemental material for: "To ensure the complete data sample was representative of the full sample, we contrasted the dataset used in the cluster analysis and the dataset of individuals with missing data using Chi-square tests for categorical measures and Mann-Whitney-U tests for continuous measures."Limitations: Mention the use of a complete case analysis, excluding those with missing risk factors, and the incomplete 12-month survey data in 59% of eligible patients. For the latter, the response rate was quite low (41%) and this may have led to a selected sample.Please submit your revised manuscript by Aug 21 2022 11:59PM. If you will need more time than this to complete your revisions, please reply to this message or contact the journal office at plosone@plos.org. Please include the following items when submitting your revised manuscript:A rebuttal letter that responds to each point raised by the academic editor and reviewer(s). You should upload this letter as a separate file labeled 'Response to Reviewers'.A marked-up copy of your manuscript that highlights changes made to the original version. You should upload this as a separate file labeled 'Revised Manuscript with Track Changes'.An unmarked version of your revised paper without tracked changes. You should upload this as a separate file labeled 'Manuscript'.

We look forward to receiving your revised manuscript.

Kind regards,

Bart Ferket

Academic Editor

PLOS ONE

Journal Requirements:

a) Did participants provide their written or verbal informed consent to participate in this study?

“Potential competing interest: JAF receives funding for the following COVID-19-related grants: NIH/NINDS 3U24NS11384401S1, NIH/NHLBI 1OT2HL161847-01, NIH/NIA 3P30AG066512-01; LET, NMS and LJB receive funding for the following COVID-19-related grant: NIH/NHLBI 1OT2HL161847-01; and TW receives funding for the following COVID-19-related grant:  NIH/NIA 3P30AG066512-01.  AdH, SY, SS, DY, AL, KM, and SLG do not report any relevant conflicts of interest.”

Reviewers' comments:

Reviewer's Responses to Questions

**Comments to the Author**

1. Is the manuscript technically sound, and do the data support the conclusions?

Reviewer #1: Yes

Reviewer #2: Yes

2. Has the statistical analysis been performed appropriately and rigorously? 

Reviewer #1: Yes

Reviewer #2: Yes

3. Have the authors made all data underlying the findings in their manuscript fully available?

Reviewer #1: Yes

Reviewer #2: Yes

4. Is the manuscript presented in an intelligible fashion and written in standard English?

Reviewer #1: Yes

Reviewer #2: Yes

5. Review Comments to the Author

Reviewer #1: I read this manuscript with a lot of interest as the subject is one that is currently not well understood with regards to the COVID-19 pandemic . The paper presents very interesting results regarding outcomes of SARS-CoV-2 infection and the response to treatment. This study is among few studies that have attempted to explore the clusters of post-Covid condition especially the treatment outcomes.

I have few suggestions to make the paper more valuable;

1. In the race to develop a case definition for long Covid, there is a growing interest in its symptom clusters. Accordingly, provide the symptoms that characterises each cluster you identified in your study (you did this for cluster 2 and 3 but not for cluster 1). Reporting that cluster 1 had few symptoms is not enough.

2. Report vaccinationation status in Table 1, preferably by number of doses. COVID-19 vaccination has been demonstrated to produce favorable outcomes. No such findings was demonstrated when adjusting for vaccination status in your study. You need to check this for this especially by dosage of COVID-19 vaccine and not vaccination status.

3. The quality of images are poor and needs a lot of improvement. Increase image quality to more than 300 dpi.

Reviewer #2: This interesting paper is about different subgroups/phenotypes of post-acute sequele of Covid-19. The authors performed unsupervised cluster analysis in order to identify subgroups that may help to management/treatment of long term symptom following the infection.

The study was based on the cohort of patients collected in the prospective manner. There were 242 patients whereas 50% reported at least one long term symptom after Covid-19 at 12 months follow-up. The study identified three cluster with few remining symptoms, with several symptoms and with specific symptoms, such as shortness of breath, headache and cognitive symptoms. The cluster with several symptoms were the most affected as they had the highest score of disability. All patients in this groups got psychiatric therapy and the major part received also physical/occupational therapy.

The title is appropriate and well inform on the study.

The abstract is very well written and contain the most important information.

The introduction is clear and consistent; however, I would suggest it can be completed with a paragraph on the previous study on subgroups/clusters as there are several studies on it already in the literature.

The method a well written. The detailed description of the cohort was published before.

-What kind of psychiatric intervations did the patients received? Please add information.

-What kind of occupational/physiotherapy did the person reciecived?

-Did you ask about in the survey and did you revive the patients records for these informations?

The results.

- I would suggest you divide the results part in the section, then it will be easier to follow-up the results text.

- I would suggest to add the simple figure with for example circles where you first have three circles of clusters (divided by symptoms) and then on them you have three circles regarding therapy.

- In the table 2 caputure, you should describe what all the rows of numbers means in each cell. As the groups are small you might consider write only OR with 95% CI and p and the numbers in the supplementary materials.

-there are several types of cluster analysis, discuss why did you use the one you selected.

The discussion:

-Is this correct that the study is the first on post covid phenotypes corresponging to the theraputic strategies? The majority of references in the paper is from 2020-2021 and you should add some more new reference and refered also on some recent paper on the phenotypes.

- You write: ‘’ Although some studies have suggested that PASC may be more common in those with mild

disease(28), we did not have a comparison group of non-hospitalized patients in our cohort to test this hypothesis.’’ Is this write with the current stage of knowladge? The patients after more severe Covid-19 used to have more somatic remaining symptoms whereas the patients after mild Covid-19 used to have cognitive impairment and fatigue.

I suggest the major revision of this paper and in particularly suggest to review the recent literature on post Covid phenotypes and complete with the references from the current year 2022. You should also mention some more other studies on cluster analysis in patients with long term symptom after Covid-19.

Thank you to letting me review this paper and good lack!

6. PLOS authors have the option to publish the peer review history of their article (what does this mean?). If published, this will include your full peer review and any attached files.

Reviewer #1: No

Reviewer #2: No

---

## [Author Response · Author response to Decision Letter 0]

9 Aug 2022

Response to Editor and Reviewers

Editor Comments:

1. Add some description of cluster 1 symptom rates in the results section based on Fig 2A (and suppl Fig 1, also see Reviewer 1 comment)

We have modified the abstract as follows (page 2, results):

“Cluster analysis generated three symptom groups: Cluster1 had few symptoms (most commonly headache)”

We have modified the results section as follows (page 6, paragraph 3, and page 7, paragraph 2)

“Hierarchical cluster analysis generated three symptom clusters (Figure 2 A and B): Symptom Cluster 1 (N=38, 31%) included patients with few symptoms, most often headache and anxiety though at lower rates than reported in other clusters; Symptom Cluster 2 (N=16, 13%) consisted of patients with many symptoms, including high levels of anxiety and depression; and Symptom Cluster 3 (N=68, 56%) included patients who predominantly reported shortness of breath, headache and cognitive abnormalities.”

“A separate hierarchical cluster analysis was performed to identify groups of patients who received similar therapeutic interventions. Three clusters were generated (Figure 3 A and B): Therapy Group A (N=72, 59%) received few therapeutic interventions, most often pain medications and exercise; Therapy Group B (N=12, 10%) received several interventions, notably psychological talk therapy, antidepressants, and anti-anxiety medications; and Therapy Group C (N=38, 31%) primarily received physical and occupational therapy.” 

2. Please redo Table 2 completely, because there are a lot of redundant data included (also see Reviewer 2 comment): 

o The N for the denominator can be given when different from given in the header or provide N (%) with missing data for applicable subsets.

o Would it make sense to assess differences in variable distributions among the Therapy groups using univariable analysis (chi squared tests and nonparametric/anova test for continuous variables)?

We appreciate these suggestions. Table 2 has been restructured to be more similar to Table 1, where we compare distributions across clusters using chi-squared and non-parametric Kruskal Wallis tests. We moved odds ratios for each demographic, comorbidity, symptom cluster and 12-month outcome variable and its relationship with each therapeutic cluster to supplemental Table 4 as suggested by Reviewer 2. 

o For the symptom clusters: it may be interesting to compare odds of falling within cluster 2 and 3 vs cluster 1 (reference group) among Therapy groups.

We appreciate this thought. We have analyzed membership in Symptom Cluster 2 or 3 versus Cluster 1 (reference) and added these data to Table 2. Patients in symptom Clusters 2/3 were more likely to be represented in Therapy Groups B and C, though this did not reach statistical significance (P=0.052). 

o For the 12-month outcomes, e.g., "improved with therapy", it may be interesting to compare Therapy group B and C vs A, while adjusting for cluster 2 and 3 dummy variables as well as severity of disease + testing for interaction between the therapy dummy variables (Therapy group B and C) and cluster dummy variables. Perhaps these analyses based on above 2 suggestions could be added separately.

The editor makes an interesting suggestion. We performed a logistic regression analysis predicting “improved with therapy” testing therapy groups B/C vs A, adjusting for covariates symptom cluster 2/3 vs 1, ventilator status (severity of index disease), age and the interaction of these symptom cluster and therapy dummy variables. However, the model was unstable, likely due to the few numbers of patients analyzed and the relatively high number of covariates. When running the model including only covariates therapy groups B/C vs A, age and ventilator status, belonging to therapy groups B/C strongly predicted subjective improvement (aOR 14.5, 95% CI 1.5-142.9, P=0.022). 

We added the following to the results section (page 8, paragraph 1):

“Conversely, belonging to Therapy Groups B or C (compared to A) was strongly associated with subjective improvement (aOR 14.5, 95% CI 1.5-142.9, P=0.022) after adjusting for age and ventilator status.”

We also examined dichotomized 12-month outcomes evaluating the variable therapy group B/C vs A adjusting for adjusting for covariates cluster 2/3 vs 1, ventilator status (severity of index disease), age and the interaction of these cluster and therapy dummy variables with the following results (all non-significant): 

For outcome mRS 4-6: therapy group B/C vs A, aOR 2.14 (0.34-13.30) P=0.415

For outcome Barthel<100: therapy group B/C vs A, aOR 4.3 (0.62-30.3) P=0.141

For tMOCA≤18: therapy group B/C vs A, aOR 0.64 (0.091-4.56) P=0.658

For anxiety NeuroQoL≥60: therapy group B/C vs A, aOR 3.06 (0.22-42.84) P=0.406

For depression NeuroQoL≥60: therapy group B/C vs A, model unstable

For sleep NeuroQoL≥60: therapy group B/C vs A, model unstable

For fatigue NeuroQoL≥60: therapy group B/C vs A, aOR 4.33 (0.22-84.9) P=0.334

Because none of the above models reached statistical significance, we did not include these in the results section.

o Were continuous variables (incl age, continuous 12-mo outcomes) analyzed as binary outcomes with the median as cut-off value? No wonder the OR are close to 1 when there are sufficient data to estimate the median, and when Ns above vs below median are compared in each cell (i.e., 0.5/(1-0.5)=1). 

For Table 2, continuous variables were entered into logistic regression to predict the dichotomous outcoming of belonging to a specific therapy group or not. We did not dichotomize continuous variables at the median. 

3. Please include results as supplemental material for: "To ensure the complete data sample was representative of the full sample, we contrasted the dataset used in the cluster analysis and the dataset of individuals with missing data using Chi-square tests for categorical measures and Mann-Whitney-U tests for continuous measures."

Of those who completed 12 month follow-up, data on post-acute symptoms was missing on 12 patients. We have added a supplemental table, as well as revised the results and discussion section to include these data.

We have added the following to the results section (page 6, paragraph 1):

“Of those who completed 12 month follow-up, data on post-acute symptoms was missing on 12 patients. When comparing data among included patients (N=122) to those with missing data (N=12), those with missing data were significantly older (median 74 versus 64 years, P=0.002), had fewer life stressors (median 0 versus 1, P=0.010) and had worse 12-month modified Rankin scores (median 4 versus 2, P<0.001, Supplemental Table 5).”

4. Limitations: Mention the use of a complete case analysis, excluding those with missing risk factors, and the incomplete 12-month survey data in 59% of eligible patients. For the latter, the response rate was quite low (41%) and this may have led to a selected sample.

We appreciate the suggestion and have added the following to the discussion limitations section (page 10, paragraph 2):

“Second, data was missing in 12 patients who were excluded from cluster analyses. Those excluded from analyses were largely similar to those included, however excluded patients were significantly older, had worse 12-month mRS scores, and had fewer life stressors than those included. Additionally, we were only able to contact 242/590 (41%) patients eligible for 12-month follow-up due to defunct contact information, language barriers, lack of patient/surrogate consent or unanswered phone calls. Though we made three phone attempts at contact, our data may represent a biased sample.”

Reviewer #1: I read this manuscript with a lot of interest as the subject is one that is currently not well understood with regards to the COVID-19 pandemic . The paper presents very interesting results regarding outcomes of SARS-CoV-2 infection and the response to treatment. This study is among few studies that have attempted to explore the clusters of post-Covid condition especially the treatment outcomes.

I have few suggestions to make the paper more valuable;

1. In the race to develop a case definition for long Covid, there is a growing interest in its symptom clusters. Accordingly, provide the symptoms that characterises each cluster you identified in your study (you did this for cluster 2 and 3 but not for cluster 1). Reporting that cluster 1 had few symptoms is not enough.

Thank you for this comment. 

We have modified the abstract as follows (page 2, results):

“Cluster analysis generated three symptom groups: Cluster1 had few symptoms (most commonly headache)”

We have modified the results section as follows (page 6, paragraph 3, and page 7, paragraph 2):

“Hierarchical cluster analysis generated three symptom clusters (Figure 2 A and B): Symptom Cluster 1 (N=38, 31%) included patients with few symptoms, most often headache and anxiety though at lower rates than reported in other clusters; Symptom Cluster 2 (N=16, 13%) consisted of patients with many symptoms, including high levels of anxiety and depression; and Symptom Cluster 3 (N=68, 56%) included patients who predominantly reported shortness of breath, headache and cognitive abnormalities.”

“A separate hierarchical cluster analysis was performed to identify groups of patients who received similar therapeutic interventions. Three clusters were generated (Figure 3 A and B): Therapy Group A (N=72, 59%) received few therapeutic interventions, most often pain medications and exercise; Therapy Group B (N=12, 10%) received several interventions, notably psychological talk therapy, antidepressants, and anti-anxiety medications; and Therapy Group C (N=38, 31%) primarily received physical and occupational therapy.” 

3. The quality of images are poor and needs a lot of improvement. Increase image quality to more than 300 dpi.

We appreciate this suggestion and have modified the figures to be 400 dpi.

Reviewer #2: This interesting paper is about different subgroups/phenotypes of post-acute sequele of Covid-19. The authors performed unsupervised cluster analysis in order to identify subgroups that may help to management/treatment of long term symptom following the infection.

The study was based on the cohort of patients collected in the prospective manner. There were 242 patients whereas 50% reported at least one long term symptom after Covid-19 at 12 months follow-up. The study identified three cluster with few remining symptoms, with several symptoms and with specific symptoms, such as shortness of breath, headache and cognitive symptoms. The cluster with several symptoms were the most affected as they had the highest score of disability. All patients in this groups got psychiatric therapy and the major part received also physical/occupational therapy.

The title is appropriate and well inform on the study.

The abstract is very well written and contain the most important information.

The introduction is clear and consistent; however, I would suggest it can be completed with a paragraph on the previous study on subgroups/clusters as there are several studies on it already in the literature.

The reviewer makes an excellent suggestion. When the paper was originally submitted in early 2022 there were not many cluster analyses evaluating post-acute long COVID symptoms. We have incorporated several references in our introduction (as below), and noted some of the differences in our study.

We have modified the introduction (page 3, paragraph 2):

“While there are cluster analyses evaluating symptoms in the acute phase of COVID-19 illness(14-17), and others focused on specific post-acute symptoms such as pulmonary complaints(18), brain fog(19), or neuropsychiatric symptoms(20), we chose to evaluate the entire post-acute COVID-19 symptom report form published by the World Health Organization (WHO)(21) and map symptom clusters to therapeutic interventions. These analyses may provide insight into the efficacy of various symptom-based treatment strategies and underscore the need for a multi-disciplinary holistic approach to post-COVID-19 care.” 

The method a well written. The detailed description of the cohort was published before.

-What kind of psychiatric intervations did the patients received? Please add information.

We have added the following to the Results section (page 7, paragraph 2):

Of 122 PASC patients, 85 (70%) received at least one therapy to manage PASC symptoms and 72/122 (59%) received multiple therapies. The most frequently prescribed interventions were physical or occupational therapy, exercise or pain medications. Psychiatric interventions included: talk therapy (9%), antidepressants (6%) and anti-anxiety medications (5%). Most patients reported a positive response to each therapy (range 75-100% reported symptom improvement), though a smaller proportion of patients (77%) felt that exercise alleviated their symptoms, and 7/85 (8%) reported no improvement with any intervention (Figure 1B.). 

-What kind of occupational/physiotherapy did the person reciecived?

The reviewer bring up an important query, however, we do not have details on the specific types of occupational and physical therapy utilized. 

-Did you ask about in the survey and did you revive the patients records for these informations?

We obtained information about therapies used directly from the patient or surrogate because many patients in the NYC area seek care at a variety of institutions and some of the therapies we were studying are not well captured in the EMR (e.g. exercise, pain meds, meditation, acupuncture, chiropracter etc). 

The results.

- I would suggest you divide the results part in the section, then it will be easier to follow-up the results text.

We appreciate this suggestion and have divided the results into sections (page 6, paragraph 1 to page 8 paragraph 3)

- I would suggest to add the simple figure with for example circles where you first have three circles of clusters (divided by symptoms) and then on them you have three circles regarding therapy.

We agree with this recommendation. A Venn diagram was difficult to construct, but we did create a figure which summarizes the relationship between symptom clusters and therapy groups (Figure 4).

- In the table 2 caputure, you should describe what all the rows of numbers means in each cell. As the groups are small you might consider write only OR with 95% CI and p and the numbers in the supplementary materials.

We appreciate these suggestions. Table 2 has been restructured to be more similar to Table 1, where we compare distributions across clusters using chi-squared and non-parametric Kruskal Wallis tests. We moved odds ratios for each demographic, comorbidity, symptom cluster and 12-month outcome variable and its relationship with each therapeutic cluster to supplemental Table 4 as suggested. 

-there are several types of cluster analysis, discuss why did you use the one you selected.

We have added the following to the Methods section (page 5, paragraph 2): 

“We selected unsupervised agglomerative hierarchical cluster analysis methods because they provided a flexible approach to assemble symptoms clusters (based on similarity of their taxonomic structures) without prior knowledge about any underlying relationships, and because they allowed us to generate easy-to-interpret views of the clustering structure, an advantage over some other clustering approaches. Agglomerative approaches are also efficient when datasets are smaller, such as ours. Unlike k-means and some other clustering approaches, this method does not require advanced selection of number of clusters, and smaller clusters tend to be generated, which was useful in our exploration of PASC sub-phenotypes. We used Ward’s minimum variance methods to minimizes the total within-cluster variance.”

The discussion:

-Is this correct that the study is the first on post covid phenotypes corresponging to the theraputic strategies? The majority of references in the paper is from 2020-2021 and you should add some more new reference and refered also on some recent paper on the phenotypes.

We appreciate this suggestion and have added more recent references to the introduction (as mentioned above) as well as to the discussion as below (page 9, paragraph 1):

“This study differs from other COVID-19 cluster analyses(15-20) in that, not only did we identify phenotypes of PASC symptoms, but we also mapped these clusters to corresponding therapeutic strategies and their subjective effectiveness.”

We did not identify any papers evaluating therapeutic groups, nor correlating symptom clusters with therapeutic strategies, so we feel our paper is unique in this regard.

- You write: ‘’ Although some studies have suggested that PASC may be more common in those with mild

disease(28), we did not have a comparison group of non-hospitalized patients in our cohort to test this hypothesis.’’ Is this write with the current stage of knowladge? The patients after more severe Covid-19 used to have more somatic remaining symptoms whereas the patients after mild Covid-19 used to have cognitive impairment and fatigue.

We agree that most of the literature supports a strong association of COVID severity with PASC symptoms (including our own). However, there is some conflicting data showing more PASC in patients with milder COVID. This could simply be because there are many more patients with mild COVID than those requiring hospitalization, or that patients with more severe COVID are too disabled to report their symptoms. The discussion section on this topic is as follows (page 10, paragraph 2):

“We found that COVID-19 severity, even amongst a hospitalized cohort, was an overall predictor of PASC in the group of 242 patients that completed follow-up, as has been described in multiple other cohorts(32-34). Although some studies have suggested that PASC may be more common in those with mild disease(35), we did not have a comparison group of non-hospitalized patients in our cohort to test this hypothesis.”

I suggest the major revision of this paper and in particularly suggest to review the recent literature on post Covid phenotypes and complete with the references from the current year 2022. You should also mention some more other studies on cluster analysis in patients with long term symptom after Covid-19.

We agree with this suggestion and have updated the introduction and discussion section with more recent references as mentioned above.

---

## [Decision Letter · Decision Letter 1]

30 Aug 2022

PONE-D-22-12045R1Post-Acute Sequelae of COVID-19 Symptom Phenotypes and Therapeutic Strategies:  A Prospective, Observational StudyPLOS ONE

Dear Dr. Frontera,

Thank you for submitting your revised manuscript to PLOS ONE. Please address additional concerns raised by reviewer 1.

We look forward to receiving your revised manuscript.

Kind regards,

Chong Chen

Academic Editor

PLOS ONE

Journal Requirements:

Reviewers' comments:

Reviewer's Responses to Questions

**Comments to the Author**

1. If the authors have adequately addressed your comments raised in a previous round of review and you feel that this manuscript is now acceptable for publication, you may indicate that here to bypass the “Comments to the Author” section, enter your conflict of interest statement in the “Confidential to Editor” section, and submit your "Accept" recommendation.

Reviewer #1: All comments have been addressed

Reviewer #2: All comments have been addressed

2. Is the manuscript technically sound, and do the data support the conclusions?

Reviewer #1: Yes

Reviewer #2: Yes

3. Has the statistical analysis been performed appropriately and rigorously? 

Reviewer #1: Yes

Reviewer #2: Yes

4. Have the authors made all data underlying the findings in their manuscript fully available?

Reviewer #1: Yes

Reviewer #2: Yes

5. Is the manuscript presented in an intelligible fashion and written in standard English?

Reviewer #1: Yes

Reviewer #2: Yes

6. Review Comments to the Author

Reviewer #1: The manuscript has improved with the additional inputs. A few minor revisions are recommended to improve the manuscript further.

Reviewer #2: Th authors adressed all the previous comments. The results are more clear now while reading and it is reflected in the discussion section with the updated reference list.

7. PLOS authors have the option to publish the peer review history of their article (what does this mean?). If published, this will include your full peer review and any attached files.

Reviewer #1: No

Reviewer #2: **Yes: **Marta A Kisiel

---

## [Author Response · Author response to Decision Letter 1]

1 Sep 2022

Reviewer 1: 

I appreciate the improvements the authors have made on the manuscript. I have one concern on the following paragraph (sections highlighted in yellow). 

1. Can the authors explain why the estimated aOR as reported in this paragraph lacks precision? Could this be a chance finding?

We appreciate this query. When contrasting cluster A (n=72) to B-C combined (n=50), we detected large differences, with low probability that these were due to chance, despite relatively small sample sizes. While the exact magnitude of difference may be imprecise, the likelihood of differences between these two groups is strong. Nonetheless, we agree that there is a wide confidence interval and the point estimate is imprecise. We have added the following to the limitations section of the discussion:

“Additionally, there were too few patients in certain treatment categories to make any statements regarding efficacy. While those in Therapy Group A reported lower rates of subjective improvement compared to Groups B or C, confidence intervals were wide suggesting imprecision in point estimates.” (page 12, paragraph 1)”

2. The last sentence that starts with “When ……. is incomplete!!!!

We thank the reviewer for catching this typo and have deleted it.

Subjective Improvement among Therapy Groups

Among patients who received at least one therapeutic intervention (N=85), those in Therapy Group B and C were significantly more likely to report improvement following therapy than Therapy Group A (100%, vs. 97% vs 83%, respectively, P=0.042). This was observed despite the fact that Therapy Group A had significantly better measures of activities of daily living (Barthel Index) and less severe disability (mRS) than other groups. After adjusting for severity of index COVID-19 illness (intubation status) and age, Therapy Group A was associated with significantly lower rates of subjective improvement compared to other therapy groups (adjusted OR 0.07, 95% CI 0.01-0.68, P=0.022). Conversely, belonging to Therapy Groups B or C (compared to A) was strongly associated with subjective improvement (aOR 14.5, 95% CI 1.5-142.9, P=0.022),

after adjusting for age and ventilator status. There was a significant association between the number of therapeutic interventions trialed and the subjective report of improvement following therapy (OR 3.00, 95% CI 1.22-7.35, P=0.017). When comparing 

3. Can the authors justify the adequacy of the sample size to perform a cluster analysis of therapy groups? It seems the sample size used to perform the clustering was not adequate. 

We appreciate this query. For cluster analyses, the goal is to determine whether subgroups exist within a sample vs a null hypothesis of a single group, and power is generally defined as the probability of correctly detecting that subgroups are present, which increases as a function of both ‘effect size’ (magnitude of difference between groups) and sample size. When separation between groups is large, small sample sizes are not a major concern. Studies have shown that most algorithms can perform optimally even with lower separations with a minimum sample size of 20–30 observations per subgroup (Dalmaijer, E.S., Nord, C.L. & Astle, D.E. Statistical power for cluster analysis. BMC Bioinformatics 23, 205 (2022). https://doi.org/10.1186/s12859-022-04675-1). In our case, we had sample sizes of 12-72 per cluster. When contrasting cluster A (n=72) to B-C combined (n=50), we detected large differences, with low probability that these were due to chance, despite relatively small sample sizes. While the exact magnitude of difference may be imprecise, the likelihood of differences between these two groups is strong.

We have added the above reference to the methods section (page 5, paragraph 2, reference 29)

4. The authors should report briefly on those who received no therapy.

The reviewer makes a good suggestion. We have added the following to the results section (page 8, paragraph 2):

“Among those who received no therapeutic interventions (N=37), the median age was 62 (IQR 51-72), 67% were male and 30% required mechanical ventilation during hospitalization. There were no significant differences in age or sex among those who did or did not receive a therapeutic intervention. However, those who did not receive any therapies were less likely to have required mechanical ventilation during index COVID-19 than those who received some intervention (30% versus 54%, P =0.018).”

---

## [Editor Report · Decision Letter 2]

13 Sep 2022

Post-Acute Sequelae of COVID-19 Symptom Phenotypes and Therapeutic Strategies:  A Prospective, Observational Study

PONE-D-22-12045R2

Dear Dr. Frontera,

We’re pleased to inform you that your manuscript has been judged scientifically suitable for publication and will be formally accepted for publication once it meets all outstanding technical requirements.

Kind regards,

Chong Chen

Academic Editor

PLOS ONE

---

## [Editor Report · Acceptance letter]

19 Sep 2022

PONE-D-22-12045R2 

Post-Acute Sequelae of COVID-19 Symptom Phenotypes and Therapeutic Strategies:  A Prospective, Observational Study 

Dear Dr. Frontera:

I'm pleased to inform you that your manuscript has been deemed suitable for publication in PLOS ONE. Congratulations! Your manuscript is now with our production department. 

Kind regards, 

on behalf of

Dr. Chong Chen 

Academic Editor

PLOS ONE